# Affine Formation Maneuver Control for Multi-Heterogeneous Unmanned Surface Vessels in Narrow Channel Environments

**Yeye Liu [1,*], Xiaogong Lin [1] and Chao Zhang [2]**

[1] College of Intelligent Systems Science and Engineering, Harbin Engineering University, Harbin 150001, China; linxiaogong@hrbeu.edu.cn

[2] Southwest Institute of Technical Physics, Chengdu 610046, China; mario_zhangc@126.com

\* Correspondence: lyy317040129@hrbeu.edu.cn; Tel.: +86-13206653035

**Abstract:** This paper investigates the affine formation maneuver control for multi-heterogeneous unmanned surface vessels (USV), aiming to enable them to navigate through narrow channels in the near-sea environment. The approach begins with implementing an affine transformation to facilitate flexible configuration adjustments within the formation system. The affine transformation of the entire formation is achieved by controlling the leaders' positions. Second, this article introduces an anti-perturbation formation tracking controller for the underactuated vessels, which assume the role of leaders, to accurately follow the arbitrary formation transformation. Third, the followers consist of fully actuated vessels with the same kinematic model as the leaders but different dynamic models. This paper utilizes the affine localizability theorem to derive an expected virtual time-varying trajectory based on the leaders' trajectory. The followers achieve the desired formation maneuver control by tracking this expected virtual time-varying trajectory through an anti-perturbation formation tracking controller. Finally, the efficacy of the introduced control law is confirmed and supported by the results obtained from rigorous simulation experiments.

**Keywords:** multi-heterogeneous unmanned surface vessels; disturbance estimation; formation tracking control; affine transformation

## 1. Introduction

The field of unmanned maritime vehicles stands as a beacon of technological progress. Within this realm, the importance of unmanned surface vessel (USV) formation technology is consistently growing, driven by its remarkable strides in recent years [1–4]. USV formation represents a dynamic and collaborative approach that involves the orchestrated actions of multiple unmanned surface vessels. Its vast potential extends across an impressive range of essential domains, encompassing oceanography, resource surveying, maritime traffic management, and the critical realm of rescue operations. Embracing the concept of USV formation enables the realization of an unprecedented level of intelligent coordination and seamless communication among these autonomous vessels. This achievement not only significantly enhances task execution efficiency and safety, but it also simultaneously reduces our dependency on human resources, driving down overall operational costs [5–8]. The multifaceted benefits of USV formation have garnered attention and genuinely captivated the maritime industry, captivating stakeholders with the immense value it offers. As ongoing advancements in this technology unfold, a transformative era is on the horizon, poised to reshape our approach to the myriad of challenges naval operations pose. This ushering in of a new generation brings remarkable capabilities and opportunities, underscoring the profound impact that autonomous maritime systems, driven by USV formation, will have on our naval future. The convergence of advanced technology, strategic coordination, and optimized resource utilization defines this exciting trajectory, promising a lot where innovation and efficiency harmoniously navigate the complexities of our oceans.

Control methods play a pivotal role in USV formations and can be categorized into two fundamental types: formation-keeping control and formation maneuver control [4,9,10]. The primary goal of formation-keeping control is to sustain a relatively stable configuration within the USV formation, ensuring the continuous and precise execution of formation-related tasks. The maritime environment presents USVs with intricate external disturbances, such as waves, winds, and currents, adding to the challenge. Concurrently, model uncertainties further complicate the endeavor, potentially compromising the formation's stability [3,11–14]. Consequently, formation-keeping control stands as a prominent research area in USV formations, with diverse control algorithms and methods meticulously examined to ensure the preservation of predetermined relative positions and spacing, even amidst the volatility of maritime conditions. Recent studies [15–18] have significantly contributed to the formation control problem by introducing novel algorithms, control strategies, and theoretical advancements. These papers have explored various aspects of formation control, ranging from distributed control algorithms that enhance robustness and adaptability to using advanced sensing technologies for improved precision and coordination. However, exclusive reliance on formation-keeping control may only partially meet the demands for efficient navigation and flexible maneuverability during real-world naval missions. To address the complexities and variabilities inherent in the marine environment, along with the specific mission requirements, the concept of formation maneuver control has garnered significant attention. Formation maneuver control empowers USVs to uphold their relative positions within the formation while executing rapid and secure maneuvering actions, encompassing translation, rotation, scaling, and shearing [6,19,20]. Through formation maneuver control, USV formations can swiftly respond to emergent situations, evade obstacles, and optimize route planning, thereby augmenting the overall adaptability and efficiency of formation-related tasks. This issue serves as the primary motivation driving the focus of this study, which aims to contribute to advancing USV formation technologies.

Many formation maneuver control strategies have emerged in recent years, encompassing bearing-based, displacement-based, and distance-based control laws and techniques grounded in barycentric coordinates and complex Laplacians [21–31]. These diverse control methodologies possess unique advantages and applicability, offering various options for advancing research in USV formations. Within the realm of bearing-based control, this method predominantly focuses on adjusting the relative angles between USVs within the formation, often employed for fine-tuning the orientation of vessels [27,30]. By manipulating the heading angles of the USVs, this approach facilitates formation maneuvers and adaptability to varying mission requirements and dynamic environmental conditions. However, the displacement-based and distance-based control methods modify the relative positions among the USVs within the formation [22,24–26,31]. By effectively controlling the displacements and distances between the vessels, these methods maintain specific spacing and positional relationships, thereby ensuring relative stability while allowing for flexibility to adjust the formation's shape to meet situational demands. Barycentric coordinates introduce a centroid-centric coordinate system for describing the formation's structure, enabling the manipulation of centroid coordinates to shape and optimise the formation's arrangement [32]. On the contrary, complex Laplacians employ sophisticated operators to simulate interactions and cooperative behaviors within the formation, providing more versatile and comprehensive control capabilities for shaping formations [21,29]. Despite the achievements of these control methods in formation research, substantial challenges persist, particularly in grappling with the intricacies of the marine environment and attaining efficient control over formation maneuvers. To further elevate the performance and adaptability of USV formations, the integration of affine transformation technology emerges as a promising avenue for exploration. Affine transformations confer the flexibility to fine-tune the formation's shape and structure, offering advanced and diverse control strategies for formations [28,33–36]. Consequently, this paper will focus on the comprehensive study of affine transformation technology, delving into its application within USV formations to

contribute meaningfully to the continued advancement of USV formation technology. This issue is the second motivation for this study.

Furthermore, conventional USV formation systems typically adopt a homogeneous formation, wherein all USVs share identical characteristics and control strategies [37]. While this formation type is straightforward to implement, it presents certain limitations when confronted with intricate and dynamic environmental conditions and diverse mission requirements. Homogeneous formations need help to fully adapt to the diverse demands of various tasks and miss out on leveraging the unique strengths and advantages possessed by individual USVs. Thus, to further broaden the applicability of USV formations, the incorporation of heterogeneous formations has emerged as an indispensable research avenue. Heterogeneous formations, achieved through integrating USVs with distinct characteristics and capabilities, allow for comprehensively utilising each USV's strengths, catering more effectively to diverse mission requirements. For instance, including fully actuated and underactuated USVs within a formation enables the fully actuated ones to demonstrate agile control capabilities while the underactuated counterparts excel in executing sustained tasks. Moreover, introducing leader–follower concepts enhances the flexibility and intelligence of heterogeneous formations, enabling them to accomplish more sophisticated and cooperative actions. This aspect constitutes the third primary motivation driving the focus of this study, aiming to contribute to the advancement of heterogeneous USV formation technology.

This paper explores heterogeneous formation technology, particularly in USV formations in near-sea environments. The approach encompasses utilising affine transformation, the affine localizability theorem, and an anti-perturbation formation tracking controller to efficiently navigate USV formations through narrow channels. Through this study, fresh insights and methodologies are intended to be provided for the development and application of USV formation technology, with the ultimate goal of advancing the widespread adoption of unmanned surface vessel technology in the marine domain. The key contributions can be outlined as follows:

1. The combination of fully actuated and underactuated vessels creates a novel class of heterogeneous formation systems, paving the way for further investigations into other heterogeneous stratigraphic systems;
2. This paper proposes an anti-perturbation affine formation maneuver controller for the leaders to effectively handle the challenges of offshore vessel applications and various factors influencing formation control, including model uncertainties and environmental disturbances;
3. This paper leverages the affine localizability theorem to achieve seamless formation maneuver control to derive an expected virtual time-varying trajectory based on the leaders' trajectory. Subsequently, the followers effectively realize the desired formation maneuver control by tracking this expected virtual time-varying trajectory using an anti-perturbation formation tracking controller.

The following parts of this paper are structured in the following manner. Section 2 presents the motion dynamics of fully actuated and underactuated vessels and establishes foundational notations for affine formations. The control problem is then formulated based on these dynamics and notes. Section 3 proposes two distinct formation controllers for fully actuated and underactuated vessels. Section 4 showcases the simulation results, providing evidence to validate the effectiveness of our proposed approach. Finally, Section 5 concludes this paper and glimpses potential future research directions.

## 2. Preliminaries and Problem Formulation

### 2.1. Model Description

The object of this paper is a heterogeneous formation system consisting of $N_l$ underactuated vessels and $N_f$ fully actuated vessels. The kinematics of the underactuated and fully actuated vessels are modelled identically as follows:

$$\dot{\eta}_i = \mathcal{J}(\psi_i) v_i \ i \in N_l \cup N_f, \tag{1}$$

where $\eta_i = [p_i, \psi_i]^T \in \mathbb{R}^3$ is the posture vector in the earth-fixed frame $\mathbb{E}$, $p_i = [x_i, y_i]$ denotes the position of the vessel $i$, $\psi_i$ denotes the heading angle of the vessel $i$; $v_i = [o_i, r_i]^T \in \mathbb{R}^3$ is the velocity vector in the body-fixed frame $\mathbb{B}_i$, $o_i = [u_i, v_i]$ denotes the surge and sway velocity of the vessel $i$, $r_i$ denotes the yaw rate; and $\mathcal{J}(\psi_i)$ is the rotation matrix associated with the heading angle $\psi_i$, which is as follows:

$$\mathcal{J}(\psi_i) = \begin{bmatrix} \mathcal{R}(\psi_i) & 0_{2\times 1} \\ 0_{1\times 2} & 1_{1\times 1} \end{bmatrix}, \quad \mathcal{R}(\psi_i) = \begin{bmatrix} \cos\psi_i & -\sin\psi_i \\ \sin\psi_i & \cos\psi_i \end{bmatrix} \tag{2}$$

The dynamics of the underactuated vessel $i$ are modelled as follows:

$$\begin{cases} \dot{u}_i = f_u^i + \frac{\tau_{wu}^i}{m_{11}^i} + \frac{\tau_u^i}{m_{11}^i} \\ \dot{v}_i = f_v^i + \frac{\tau_{wv}^i}{m_{22}^i} & i \in N_l, \\ \dot{r}_i = f_r^i + \frac{\tau_{wr}^i}{m_{33}^i} + \frac{\tau_r^i}{m_{33}^i} \end{cases} \tag{3}$$

where $f_u^i = -\frac{1}{m_{11}^i}\left(c_{13}^i r_i + d_{11}^i u_i + g_u^i\right)$, $f_v^i = -\frac{1}{m_{22}^i}\left(c_{23}^i r_i + d_{22}^i v_i + d_{23}^i r_i + g_v^i\right)$, $f_r^i = -\frac{1}{m_{33}^i}\left(c_{31}^i u_i + c_{32}^i v_i + d_{32}^i v_i + d_{33}^i r_i + g_r^i\right)$; $m_{11}^i$, $m_{22}^i$, and $m_{33}^i$ represent the inertia coefficient including added mass effects. The variables $c_{13}^i$, $c_{23}^i$, $c_{31}^i$, and $c_{32}^i$ are Coriolis and centripetal coefficients; $d_{11}^i$, $d_{22}^i$, $d_{23}^i$, $d_{32}^i$, and $d_{33}^i$ are hydrodynamic damping coefficients; $g_u^i$, $g_v^i$, and $g_r^i$ are uncertain dynamics; $\tau_{wu}^i$, $\tau_{wv}^i$, and $\tau_{wr}^i$ are forces or moments of external time-varying disturbances, such as wind, waves, and currents; $\tau_u^i$ and $\tau_r^i$ are actuator inputs for the vessel $i$.

The dynamics of the fully actuated vessel $i$ are modelled as follows:

$$M_i \dot{v}_i + C_i v_i + D_i v_i = \tau_i + \tau_w^i \ i \in N_f, \tag{4}$$

where $\tau_i \in \mathbb{R}^3$ denotes the control vector; $\tau_w^i \in \mathbb{R}^3$ denotes the environmental vector; $M_i$ denotes the inertial matrix; $C_i$ denotes the Coriolis and centripetal matrix; and $D_i$ denotes the hydrodynamic damping matrix. These matrices are given as:

$$M_i = \begin{bmatrix} m_{11}^i & 0 & 0 \\ 0 & m_{22}^i & m_{23}^i \\ 0 & m_{32}^i & m_{33}^i \end{bmatrix}, \quad C_i = \begin{bmatrix} 0 & 0 & c_{13}^i \\ 0 & 0 & c_{23}^i \\ c_{31}^i & c_{32}^i & 0 \end{bmatrix}, \quad D_i = \begin{bmatrix} d_{11}^i & 0 & 0 \\ 0 & d_{22}^i & d_{23}^i \\ 0 & d_{32}^i & d_{33}^i \end{bmatrix}. \tag{5}$$

Before proceeding, the following assumptions are made.

**Assumption 1** ([12]). *The environmental disturbances and their derivatives are time-varying and bounded.*

**Remark 1.** *Due to the continuous changes in the external environment and its finite energy resources, the external disturbances acting on the water surface vessel are characterized by their unknown, time-varying, and bounded nature. Therefore, Assumption 3.1 is justified.*

**Assumption 2** ([33]). *All leaders' velocities and derivatives are assumed to be smooth and upper-bounded by known limits.*

**Remark 2.** *The desired formation trajectory is planned using the polynomial trajectory planning method, which involves interpolating a sequence of carefully chosen reference points using polynomials. Therefore, Assumption 2 is justified. For further details, please refer to* [20].

**Assumption 3.** *The leader's velocity and velocity derivative are assumed to be measurable, and the upper bounds of disturbances and the derivatives of disturbances are known.*

### 2.2. Definitions for Affine Transformation

First, an undirected graph $\mathcal{G} = \{\mathcal{V}, \mathcal{E}\}$ is employed to describe the communication topology in the heterogeneous formation system, which consists of a node set $\mathcal{V} =$ describe the set of neighbors for the vessel $i$. Let $\Gamma$ be a nominal configuration associated with the graph $\mathcal{G}$, which can be expressed as $\Gamma = \left[\Gamma_l, \Gamma_f\right]^T = [\gamma_1, \gamma_2, \cdots, \gamma_N]^T$. Next, some necessary definitions and lemmas about affine transformation are presented.

**Definitions 1** ([35]). *For formation $(\mathcal{G}, \Gamma)$ , let $w_{ij} \in \mathbb{R}$ be a stresatisfiesessponding edge $(i, j) \in \mathcal{E}$. The stress is a scalar and satisfied by $w_{ij} = w_{ji}$. If $\sum w_{ij}(\gamma_j - \gamma_i) = 0$ for all $i \in \mathcal{V}$ and $j \in \mathcal{N}_i$, the set $\Pi$, consisting of $w_{ij}$, is considered as an equilibrium stress set and $\Pi$ satisfies the following equality:*

$$(\Pi \otimes I_2)\Gamma = 0, \tag{6}$$

*where $\Pi \in \mathbb{R}^{N \times N}$, satisfying:*

$$[\Pi]_{ij} = \begin{cases} 0 & i \neq j, (i, j) \in E \\ -w_{ij} & i \neq j, (i, j) \in E \\ \sum_{k \in \mathcal{N}_i} w_{ik} & i = j \end{cases} \tag{7}$$

**Definitions 2** ([35]). *For formation $(\mathcal{G}, \Gamma)$ , the affine span of the nominal configuration $\{\gamma_i\}_{i=1}^N$ is as follows:*

$$\mathcal{A} = \left\{ \sum_{i=1}^N \alpha_i \gamma_i \middle| \gamma_i \in \mathbb{R}^2, \ \alpha_i \in \mathbb{R} \ and \ \sum_{i=1}^N \alpha_i = 1 \right\} \tag{8}$$

**Definitions 3** ([35]). *The affine image is a set consisting of all the affine transformations of the normal configuration $\{\gamma_i\}_{i=1}^N$ and defined as follows:*

$$\mathcal{S}(\Gamma) = \left\{ \mathcal{P} = \left[p_1^T, p_2^T, \cdots, p_N^T\right]^T \in \mathbb{R}^{2N} \middle| p_i = A\gamma_i + b \right\}, \tag{9}$$

*where $A \in \mathbb{R}^{2 \times 2}$ and $b \in \mathbb{R}^2$ are the affine transformation.*

**Definitions 4** ([35]). *For the formation, $(\mathcal{G}, \Gamma)$ is affinely localizable if both of the following conditions are satisfied: i. $\{\gamma_i\}_{i=1}^N$ affinely span $\mathbb{R}^2$; ii. Any $P = \left[p_l^T, p_f^T\right]^T \in \mathcal{A}(\Gamma)$, i.e., $p_f$ can be uniquely determined by $p_l$.*

**Lemma 1** ([35]). *Given an augmented matrix $\overline{\Gamma} = [\Gamma, \mathbf{1}_N] \in \mathbb{R}^{N \times 3}$ , the normal configuration $\{\gamma_i\}_{i=1}^N$ affinely span $\mathbb{R}^2$ if and only if $N \geq 3$ and $rank(\overline{\Gamma}) = 3$.*

**Lemma 2** ([35]). *The normal configuration $(\mathcal{G}, \Gamma)$ is universally rigid if and only if there exists a stress matrix $\Pi$ such that $\Pi$ is positive semi-definite and $rank(\Gamma) = N - 3$.*

**Lemma 3** ([35]). *In the condition of $\{\gamma_i\}_{i=1}^N$ affinely span $\mathbb{R}^2$, the normal configuration $(\mathcal{G}, \Gamma)$ is affinely localizable if and only if $\{\gamma_i\}_{i \in N_l}$ affinely span $\mathbb{R}^2$.*

**Lemma 4** ([35]). *For the stress matrix, denote:*

$$\overline{\Pi} = \Pi \otimes I_2 = \begin{bmatrix} \overline{\Pi}_{ll} & \overline{\Pi}_{lf} \\ \overline{\Pi}_{fl} & \overline{\Pi}_{ff} \end{bmatrix}, \tag{10}$$

*with $\overline{\Pi}_{fl} \in \mathbb{R}^{2N_f \times 2N_l}$ and $\overline{\Pi}_{ff} \in \mathbb{R}^{2N_f \times 2N_f}$. If the normal configuration $\{\gamma_i\}_{i=1}^N$ affinely span $\mathbb{R}^2$ and has a positive semi-definite stress matrix $\Pi$, then it is affinely localizable if and only if $\overline{\Pi}_{ff}$ is nonsingular. When $\overline{\Pi}_{ff}$ is nonsingular, for any $\left[ p_l^T, p_f^T \right]^T \in \mathcal{A}(\Gamma)$, $p_f$ can be uniquely calculated, and $p_f = -\overline{\Pi}_{ff}^{-1} \overline{\Pi}_{fl} p_l$.*

### 2.3. Control Objective

The primary aim of this paper is to ensure the convergence of all vessels to their designated positions, thereby achieving a target formation through coordinated maneuvers within challenging narrow channel environments, i.e.,

$$\begin{aligned} &\lim_{t \to \infty} \left( p_l(t) - p_l^*(t) \right) = 0 \\ &\lim_{t \to \infty} \left( p_f(t) - p_f^*(t) \right) = \lim_{t \to \infty} \left( p_f(t) + \overline{\Pi}_{ff}^{-1} \overline{\Pi}_{fl} p_l(t) \right) = 0. \end{aligned} \tag{11}$$

## 3. Affine Formation Maneuver Control Design

This section employs two practical control algorithms for leaders and followers to successfully attain the desired formation maneuver control. The control framework for this formation maneuver is visually illustrated in Figure 1.

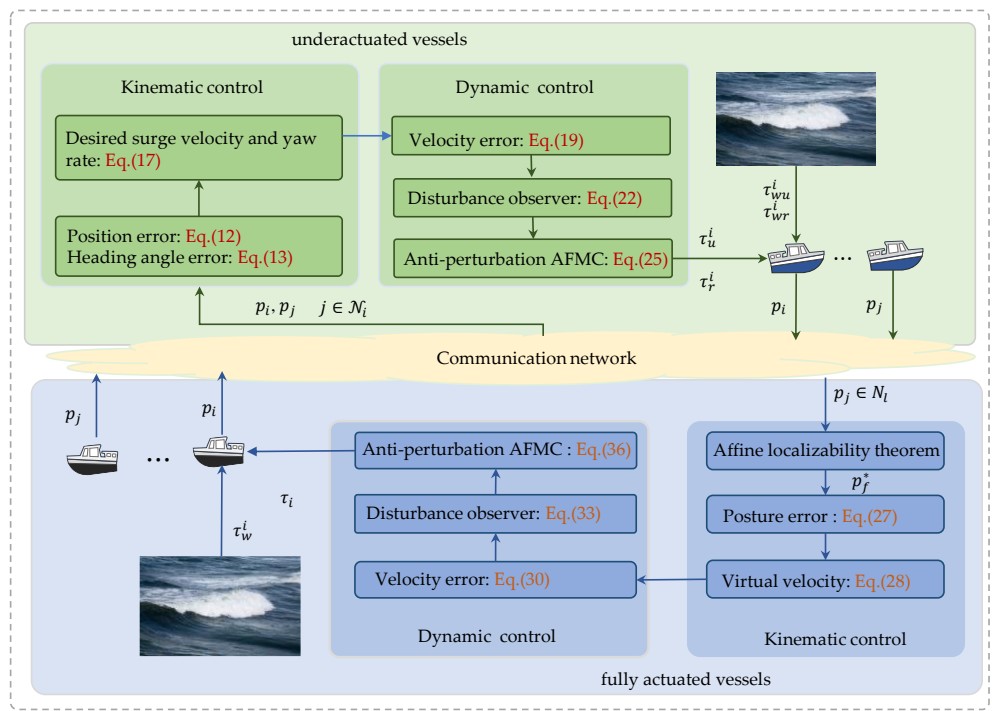

**Figure 1.** The control framework in Section 3.

### 3.1. Formation Tracking Control Design for the Leaders

Considering that the time-varying maneuver parameters $A(t)$ and $b(t)$ are decided by the leaders, the objective is to design a tracking control such that $\lim_{t \to \infty} \left( p_l(t) - p_l^*(t) \right) = 0$. The details of the control design are provided in the following three steps.

**Remark 3.** *The intended trajectory for the formation is established through a polynomial trajectory planning approach, which entails interpolating a well-selected sequence of reference points using polynomial functions. Thus, the validity of the previously mentioned assumption is substantiated. For more comprehensive information, please consult the work of* [20].

Step 1: Define the position error as follows:

$$\varkappa_1^i = \|p_i - p_i^*\|_2. \tag{12}$$

Define the heading angle error as follows:

$$\varkappa_2^i = \psi_i - \psi_i^*, \tag{13}$$

where $\psi_i^*$ is the desired orientation angle for the $i$th leader, which has the following form:

$$\psi_i^* = \begin{cases} 0.5\left(1 - sign\left(\widetilde{x}_i\right)\right)sign\left(\widetilde{y}_i\right)\pi + \arctan\left(\widetilde{y}_i/\widetilde{x}_i\right), & \varkappa_1^i \neq 0 \\ \mathrm{atan2}\left(y_i^*/x_i^*\right), & \varkappa_1^i = 0 \end{cases}, \tag{14}$$

where $\widetilde{x}_i = x_i - x_i^*$, $\widetilde{y}_i = y_i - y_i^*$.

Derivation of Equation (12) yields:

$$\dot{\varkappa}_1^i = u_i\cos\left(\varkappa_2^i\right) + v_i\sin\left(\varkappa_2^i\right) - \dot{x}_i^*\cos(\psi_i^*) - \dot{y}_i^*\sin(\psi_i^*). \tag{15}$$

Derivation of Equation (13) yields:

$$\dot{\varkappa}_2^i = r_i - \dot{\psi}_i^*. \tag{16}$$

The virtual control law ensures the leader can reach the desired position:

$$\begin{aligned} u_d^i &= \left(\cos\left(\varkappa_2^i\right)\right)^{-1}\left(-k_u^i\varkappa_1^i + v_i\sin\left(\varkappa_2^i\right) + \dot{x}_i^*\cos\left(\psi_i^*\right) + \dot{y}_i^*\sin\left(\psi_i^*\right)\right) \\ r_d^i &= -k_r^i\varkappa_2^i + \dot{\psi}_i^* \end{aligned}, \tag{17}$$

where $k_u^i$ and $k_r^i$ are positive parameters.

Step 2: To avoid derivation of the desired velocity, let (17) pass through the following first-order filter:

$$\begin{aligned} T_u^i\dot{u}_d^i &= u_d^i - u_d^i \\ T_r^i\dot{r}_d^i &= r_d^i - r_d^i \end{aligned}, \tag{18}$$

where $T_u^i$ and $T_r^i$ are time constants. Define the velocity errors as follows:

$$\begin{aligned} e_1^i &= u_i - u_d^i \\ e_2^i &= r_i - r_d^i \\ e_3^i &= u_d^i - u_d^i \\ e_4^i &= r_d^i - r_d^i. \end{aligned} \tag{19}$$

Derivation of Equation (19) yields:

$$\begin{aligned} \dot{e}_1^i &= f_u^i + \frac{\tau_{wu}^i}{m_{11}^i} + \frac{\tau_u^i}{m_{11}^i} - \frac{e_3^i}{T_u^i} \\ \dot{e}_2^i &= f_r^i + \frac{\tau_{wr}^i}{m_{33}^i} + \frac{\tau_r^i}{m_{33}^i} - \frac{e_4^i}{T_r^i} \\ \dot{e}_3^i &= -\frac{e_3^i}{T_u^i} - \Lambda_r^i\left(\dot{x}_i^*, \ddot{x}_i^*, \dot{y}_i^*, \ddot{y}_i^*, \psi_i^*, \dot{\psi}_i^*, \varkappa_1^i, \dot{\varkappa}_1^i, \varkappa_2^i, \dot{\varkappa}_2^i, v_i, \dot{v}_i\right) \\ \dot{e}_4^i &= -\frac{e_4^i}{T_r^i} - \Lambda_r^i\left(\psi_i^*, \dot{\psi}_i^*, \varkappa_2^i, \dot{\varkappa}_2^i\right). \end{aligned} \tag{20}$$

$\Lambda_u^i$ and $\Lambda_r^i$ are continuous functions. The affine formation tracking control is proposed as follows:

$$\begin{array}{l} \tau_u^i = m_{11}^i\left(-\Bbbk_u^i e_1^i - e_3^i/T_u^i - \cos\left(\varkappa_2^i\right)\varkappa_1^i - \sigma_u^i\right) \\ \tau_r^i = m_{33}^i\left(-\Bbbk_r^i e_2^i - e_4^i/T_r^i - \varkappa_2^i - \sigma_r^i\right) \end{array}, \tag{21}$$

where $\Bbbk_u^i$ and $\Bbbk_r^i$ are positive parameters; $\sigma_u^i = f_u^i + \frac{\tau_{wu}^i}{m_{11}^i}$ and $\sigma_r^i = f_r^i + \frac{\tau_{wr}^i}{m_{33}^i}$ are lumped disturbances consist of environmental disturbances and model uncertainties, which will be composited by the disturbance observer in step 3.

Step 3: Let $\chi_i = \left[u_i.v_i, r_i, \sigma_u^i, \sigma_v^i, \sigma_r^i\right]^T$ and $\mathcal{U}_i = \left[\frac{\tau_u^i}{m_{11}^i}, 0, \frac{\tau_r^i}{m_{33}^i}, 0, 0, 0\right]^T$, and rewrite (3) in state-space form, which derives:

$$\dot{\chi}_i = \mathcal{B}_i\chi_i + \mathcal{U}_i + \mathcal{Q}_i h_i, \tag{22}$$

where $\mathcal{B}_i = \begin{bmatrix} 0_{3\times3} & I_{3\times3} \\ 0_{3\times3} & 0_{3\times3} \end{bmatrix}$, $\mathcal{Q}_i = \begin{bmatrix} 0_{3\times3} \\ I_{3\times3} \end{bmatrix}$, $h_i = \left[\dot{\sigma}_u^i, 0, \dot{\sigma}_r^i\right]^T$. The observer is proposed:

$$\dot{\hat{\chi}}_i = \mathcal{B}_i\hat{\chi}_i + \mathcal{U}_i + \mathcal{H}_i\mathcal{F}_i\tilde{\chi}_i, \tag{23}$$

where $\hat{\chi}_i = \left[\hat{u}_i, \hat{v}_i, \hat{r}_i, \hat{\sigma}_u^i, \hat{\sigma}_v^i, \hat{\sigma}_r^i\right]^T$ is the estimation of $\chi_i$, $\tilde{\chi}_i = \chi_i - \hat{\chi}_i$ is the estimation error, $\mathcal{F}_i = \left[I_{3\times3} \quad 0_{3\times3}\right]$, and $\mathcal{H}_i \in \mathbb{R}^{6\times3}$ is the constant observer gain, which is proposed to be computed as follows:

$$\mathcal{H}_i = -\mathcal{P}_i^T\Phi_i, \tag{24}$$

where $\mathcal{P}_i \in \mathbb{R}^{6\times6}$ and $\Phi_i \in \mathbb{R}^{6\times3}$ are part of the solution of the optimization problem, which can get by employing some LMIs:

$$\begin{array}{ll} \min\limits_{\mathcal{P}_i,\Phi_i,\delta_i} & \delta_i \\ s.t. & \mathcal{P}_i > 0, \\ & \mathcal{B}_i^T\mathcal{P}_i + \mathcal{P}_i\mathcal{B}_i + \mathcal{F}_i^T\Phi_i^T + \Phi_i\mathcal{F}_i + I \leq 0 \\ & \begin{bmatrix} -\delta_i I & \mathcal{P}_i\mathcal{Q}_i \\ * & -\delta_i \end{bmatrix} \leq 0 \end{array}, \tag{25}$$

where $\delta_i \in \mathbb{R}$ is the decision variable, then the affine formation tracking control with the disturbance estimation is proposed as follows:

$$\begin{array}{l} \tau_u^i = m_{11}^i\left(-\Bbbk_u^i e_1^i - \dfrac{e_3^i}{T_u^i} - \cos\left(\varkappa_2^i\right)\varkappa_1^i - \hat{\sigma}_u^i\right) \\ \tau_r^i = m_{33}^i\left(-\Bbbk_r^i e_2^i - \dfrac{e_4^i}{T_r^i} - \varkappa_2^i - \hat{\sigma}_r^i\right) \end{array}, \tag{26}$$

where $\hat{\sigma}_u^i$ and $\hat{\sigma}_r^i$ are the fourth and sixth elements in $\hat{\chi}_i$.

### 3.2. Formation Tracking Control Design for the Followers

With (20) in play, the leaders can accurately track the desired positions. In this subsection, control algorithms are devised for the followers to achieve the objective of $\lim\limits_{t\to\infty}\left(p_f(t) + \overline{\Pi}_{ff}^{-1}\overline{\Pi}_{fl}p_l(t)\right) = 0$. The details of the control design are provided in the following two steps.

Step 1: Define $p_f = \left[p_1, p_2, \cdots, p_{N_f}\right]^T$, $\overline{\Pi}_{ff}^{-1}\overline{\Pi}_{fl}p_l = \left[p_1^*, p_2^*, \cdots, p_{N_f}^*\right]^T$. The desired heading angle is as follows:

$$\psi_i^* = \text{atan2}(y_i^*, x_i^*). \tag{27}$$

Let $\eta_i^* = \left[ (p_i^*)^T, \psi_i^* \right]^T$, the position error as follows:

$$\varkappa_3^i = \eta_i - \eta_i^*. \tag{28}$$

The desired virtual control law is as follows:

$$\varsigma_i = \mathcal{J}^T(\psi_i)\left(-K_1^i \varkappa_3^i + \dot{\eta}_i^*\right), \tag{29}$$

where $K_1^i \in \mathbb{R}^{3\times3}$ is the gain matrix.

Step 2: To avoid complex differentiation and simplify controller design, let $\varsigma_i$ pass through a first-order filter:

$$T_\ell^i \dot{\varsigma}_i = \varsigma_i - \varsigma_i. \tag{30}$$

where $T_\ell^i \in \mathbb{R}^{3\times3}$ is the time constant matrix. Define the velocity errors as follows:

$$\begin{aligned} e_5^i &= v_i - \varsigma_i \\ e_6^i &= \varsigma_i - \dot{\varsigma}_i \end{aligned} \tag{31}$$

The affine formation tracking control is proposed as follows:

$$\tau_i = M_i\left(-K_2^i e_5^i - \mathcal{J}^T(\psi_i)\varkappa_3^i - e_6^i/T_\ell^i - \sigma_i\right). \tag{32}$$

where $K_2^i \in \mathbb{R}^{3\times3}$ is the gain matrix; $\sigma_i = \tau_w^i - C_i v_i - D_i v_i$ is lumped disturbances that consist of environmental disturbances and model uncertainties, which will be composited by the disturbance observer in Step 3.

Step 3: Let $\chi_i = \left[v_i^T, \sigma_i^T\right]^T$ and $\mathcal{U}_i = M_i^{-1}\tau_i$, and rewrite (4) in state-space form, which derives:

$$\dot{\chi}_i = \mathcal{B}_i \chi_i + \mathcal{U}_i + \mathcal{Q}_i h_i, \tag{33}$$

where $\mathcal{B}_i = \begin{bmatrix} 0_{3\times3} & I_{3\times3} \\ 0_{3\times3} & 0_{3\times3} \end{bmatrix}$, $\mathcal{Q}_i = \begin{bmatrix} 0_{3\times3} \\ I_{3\times3} \end{bmatrix}$, $h_i = \dot{\sigma}_i$. The observer is proposed:

$$\dot{\hat{\chi}}_i = \mathcal{B}_i \hat{\chi}_i + \mathcal{U}_i + \mathcal{H}_i \mathcal{F}_i \tilde{\chi}_i, \tag{34}$$

where $\hat{\chi}_i = \left[\hat{v}_i^T, \hat{\sigma}_i^T\right]^T$ is the estimation of $\chi_i$, $\tilde{\chi}_i = \chi_i - \hat{\chi}_i$ is the estimation error, $\mathcal{F}_i = \begin{bmatrix} I_{3\times3} & 0_{3\times3} \end{bmatrix}$, and $\mathcal{H}_i \in \mathbb{R}^{6\times3}$ is the constant observer gain, which is proposed to be computed as follows:

$$\mathcal{H}_i = -\mathcal{W}_i^T Y_i, \tag{35}$$

where $\mathcal{W}_i \in \mathbb{R}^{6\times6}$ and $Y_i \in \mathbb{R}^{6\times3}$ are part of the solution of the optimization problem, which can be derived by employing some LMIs:

$$\begin{aligned} &\min_{\mathcal{P}_i, \Phi_i, \epsilon_i} \epsilon_i \\ s.t. \quad & \mathcal{W}_i > 0, \\ & \mathcal{B}_i^T \mathcal{W}_i + \mathcal{W}_i \mathcal{B}_i + \mathcal{F}_i^T Y_i^T + Y_i \mathcal{F}_i + I \leq 0 \\ & \begin{bmatrix} -\epsilon_i I & \mathcal{W}_i \mathcal{Q}_i \\ * & -\epsilon_i I \end{bmatrix} \leq 0 \end{aligned} \tag{36}$$

where $\epsilon_i \in \mathbb{R}$ is the decision variable. By employing disturbance observer (34) to estimate lumped disturbances composed of model uncertainties and environmental disturbances, the affine formation tracking control with the disturbance estimation for countering the effects of disturbances is proposed as follows:

$$\tau_i = M_i\left(-K_2^i e_5^i - \mathcal{J}^T(\psi_i)\varkappa_3^i - e_6^i/T_\ell^i - \hat{\sigma}_i\right). \tag{37}$$

## 4. Stability Analysis

**Theorem 1.** *Under Assumptions 1–2, consider the underactuated vessel formation system (1) and (3). The anti-perturbation affine formation maneuver control scheme is proposed by integrating the formation tracking controller (26) and the disturbance observer (23). In that case, all the signals of the closed-loop system are bounded.*

**Proof of Theorem 1.** The candidate Lyapunov function is selected as:

$$V_1^i = \frac{1}{2}\tilde{\chi}_i^T P_i \tilde{\chi}_i. \tag{38}$$

Taking the derivative of $\tilde{\chi}_i$, one derives:

$$\dot{\tilde{\chi}}_i = (\mathcal{B}_i - \mathcal{H}_i \mathcal{F}_i)\tilde{\chi}_i + \mathcal{Q}_i h_i. \tag{39}$$

Combining the above Equations, the derivative of (38) is:

$$\dot{V}_1^i = \frac{1}{2}\tilde{\chi}_i^T \left( (\mathcal{B}_i - \mathcal{H}_i \mathcal{F}_i)^T \mathcal{P}_i + \mathcal{P}_i(\mathcal{B}_i - \mathcal{H}_i \mathcal{F}_i) \right)\tilde{\chi}_i + \tilde{\chi}_i^T \mathcal{P}_i \mathcal{Q}_i h_i. \tag{40}$$

Let $\Phi_i = -\mathcal{P}_i \mathcal{H}_i$, and rewrite (40) as:

$$\dot{V}_1^i = \frac{1}{2}\tilde{\chi}_i^T \left( \mathcal{B}_i^T \mathcal{P}_i + \mathcal{P}_i \mathcal{B}_i + \mathcal{F}_i^T \Phi_i^T + \Phi_i \mathcal{F}_i \right)\tilde{\chi}_i + \tilde{\chi}_i^T \mathcal{P}_i \mathcal{Q}_i h_i. \tag{41}$$

If the third inequality in (25) is satisfied, one derives:

$$\begin{aligned} \dot{V}_1^i & \leq -\frac{1}{2}\left\|\tilde{\chi}_i\right\|^2 + \left\|\tilde{\chi}_i\right\|\|\mathcal{P}_i\mathcal{Q}_i\|\|h_i\| \\ & \leq -\frac{1}{2}(1-\theta_i)\left\|\tilde{\chi}_i\right\|^2 \end{aligned}, \tag{42}$$

where $0 < \theta_i < 1$, and $\left\|\tilde{\chi}_i\right\| \geq \frac{2\|\mathcal{P}_i\mathcal{Q}_i\|\|h_i\|}{\theta_i}$. According to Assumption 1, one derives:

$$\left\|\tilde{\chi}_i\right\|(t) \leq \max\left\{ \Omega\left( \left\|\tilde{\chi}_i(0)\right\|, t \right), \Xi(\|h_i\|) \right\}, \tag{43}$$

where $\Omega(\bullet)$ is the $\mathscr{K}\mathscr{L}$ function, and $\Xi(\bullet)$ is the $\mathscr{K}_\infty$ function defined as:

$$\Xi(\|h_i\|) = \sqrt{\frac{\lambda_{max}(\mathcal{P}_i)}{\lambda_{min}(\mathcal{P}_i)}} \frac{2\|\mathcal{P}_i\mathcal{Q}_i\|\|h_i\|}{\theta_i}. \tag{44}$$

Hence, the estimation dynamics are ISS with respect to $h_i$.
Consider the following Lyapunov function:

$$V_2^i = \frac{1}{2}\left(\varkappa_1^i\right)^2 + \frac{1}{2}\left(\varkappa_2^i\right)^2 + \frac{1}{2}\left(e_1^i\right)^2 + \frac{1}{2}\left(e_2^i\right)^2 + \frac{1}{2}\left(e_3^i\right)^2 + \frac{1}{2}\left(e_4^i\right)^2. \tag{45}$$

Taking the derivative of $\varkappa_1^i$, $\varkappa_2^i$, $e_1^i$, $e_2^i$, $e_3^i$, and $e_4^i$, one derives:

$$
\begin{aligned}
\dot{\varkappa}_1^i &= -k_u^i \varkappa_1^i + \cos\left(\varkappa_2^i\right)\left(e_1^i + e_3^i\right) \\
\dot{\varkappa}_2^i &= -k_r^i \varkappa_2^i + e_2^i + e_4^i \\
\dot{e}_1^i &= -\frac{e_1^i}{T_u^i} - \Lambda_u^i\left(\dot{x}_i^*, \ddot{x}_i^*, \dot{y}_i^*, \ddot{y}_i^*, \psi_i^*, \dot{\psi}_i^*, \varkappa_1^i, \dot{\varkappa}_1^i, \varkappa_2^i, \dot{\varkappa}_2^i, v_i, \dot{v}_i\right) \\
\dot{e}_2^i &= -\frac{e_2^i}{T_r^i} - \Lambda_r^i\left(\psi_i^*, \dot{\psi}_i^*, \varkappa_2^i, \dot{\varkappa}_2^i\right) \\
\dot{e}_3^i &= -\mathscr{k}_u^i e_3^i - \cos\left(\varkappa_2^i\right)\varkappa_1^i + \tilde{\sigma}_u^i \\
\dot{e}_4^i &= -\mathscr{k}_r^i e_4^i - \varkappa_2^i - \tilde{\sigma}_r^i
\end{aligned}
\quad\text{,}
\tag{46}
$$

where $\tilde{\sigma}_u^i = \sigma_u^i - \hat{\sigma}_u^i$, $\tilde{\sigma}_r^i = \sigma_r^i - \hat{\sigma}_r^i$. Combining (46) and Young's inequality, the derivative of (45) is:

$$
\begin{aligned}
\dot{V}_2^i &= \varkappa_1^i \dot{\varkappa}_1^i + \varkappa_2^i \dot{\varkappa}_2^i + e_1^i \dot{e}_1^i + e_2^i \dot{e}_2^i + e_3^i \dot{e}_3^i + e_4^i \dot{e}_4^i \\
&\leq -\left(k_u^i - \frac{1}{2}\right)\left(\varkappa_1^i\right)^2 - \left(k_r^i - \frac{1}{2}\right)\left(\varkappa_2^i\right)^2 - \left(\frac{1}{T_u^i} - \frac{1}{2}\right)\left(e_1^i\right)^2 - \left(\frac{1}{T_r^i} - \frac{1}{2}\right)\left(e_2^i\right)^2 \\
&\quad -\left(\mathscr{k}_u^i - \frac{1}{2}\right)\left(e_3^i\right)^2 - \left(\mathscr{k}_r^i - \frac{1}{2}\right)\left(e_4^i\right)^2 + \Delta_i
\end{aligned}
\quad\text{,}
\tag{47}
$$

where $\Delta_i = \frac{\left|\Lambda_u^i\right|}{2} + \frac{\left|\Lambda_r^i\right|}{2} + \frac{\left|\tilde{\sigma}_u^i\right|}{2} + \frac{\left|\tilde{\sigma}_r^i\right|}{2}$ is bounded. The selection of parameters is as follows: $k_u^i > \frac{1}{2}, k_r^i > \frac{1}{2}, \mathscr{k}_u^i > \frac{1}{2}, \mathscr{k}_r^i > \frac{1}{2}, \frac{1}{T_u^i} > \frac{1}{2}, \frac{1}{T_r^i} > \frac{1}{2}$. Finally, (47) becomes:

$$
\dot{V}_2^i \leq -2\xi_i V_2^i + \Delta_i,
\tag{48}
$$

where $\xi_i = \min\left\{k_u^i - \frac{1}{2}, k_r^i - \frac{1}{2}, \frac{1}{T_u^i} - \frac{1}{2}, \frac{1}{T_r^i} - \frac{1}{2}, \mathscr{k}_u^i - \frac{1}{2}, \mathscr{k}_r^i - \frac{1}{2}\right\}$. By solving Equation (48), one derives:

$$
0 \leq V_2^i(t) \leq \frac{\Delta_i}{2\xi_i} + \left(V_2^i(0) - \frac{\Delta_i}{2\xi_i}\right)e^{-2\xi_i t},
\tag{49}
$$

which implies that $\lim_{t\to\infty} V_2^i(t) = \frac{\Delta_i}{2\xi_i}$ and all the signals of the closed-loop system are bounded. $\square$

**Theorem 2.** *Under Assumptions 1–2, consider the fully actuated vessel formation system (1) and (4). The anti-perturbation affine formation maneuver control scheme is proposed by integrating the formation tracking controller (37) and the disturbance observer (34). In that case, all the signals of the closed-loop system are bounded.*

**Proof of Theorem 2.** The candidate Lyapunov function is selected as:

$$
V_3^i = \frac{1}{2}\tilde{\chi}_i^T P_i \tilde{\chi}_i.
\tag{50}
$$

Taking the derivative of $\tilde{\chi}_i$, one derives:

$$
\dot{\tilde{\chi}}_i = (\mathcal{B}_i - \mathcal{H}_i \mathcal{F}_i)\tilde{\chi}_i + \mathcal{Q}_i h_i.
\tag{51}
$$

Combining the above Equations, the derivative of (48) is:

$$
\dot{V}_3^i = \frac{1}{2}\tilde{\chi}_i^T\left((\mathcal{B}_i - \mathcal{H}_i \mathcal{F}_i)^T \mathcal{P}_i + \mathcal{P}_i(\mathcal{B}_i - \mathcal{H}_i \mathcal{F}_i)\right)\tilde{\chi}_i + \tilde{\chi}_i^T \mathcal{P}_i \mathcal{Q}_i h_i.
\tag{52}
$$



Let $Y_i = -\mathcal{W}_i \mathcal{H}_i$, and rewrite (52) as:

$$\dot{V}_3^i = \frac{1}{2}\widetilde{\chi}_i^T \left( \mathcal{B}_i^T \mathcal{W}_i + \mathcal{W}_i \mathcal{B}_i + \mathcal{F}_i^T Y_i^T + Y_i \mathcal{F}_i \right) \widetilde{\chi}_i + \widetilde{\chi}_i^T \mathcal{W}_i \mathcal{Q}_i h_i. \tag{53}$$

If the third inequality in (37) is satisfied, one derives:

$$\begin{aligned} \dot{V}_3^i &\leq -\tfrac{1}{2}\left\|\widetilde{\chi}_i\right\|^2 + \left\|\widetilde{\chi}_i\right\| \|\mathcal{W}_i \mathcal{Q}_i\| \|h_i\| \\ &\leq -\tfrac{1}{2}(1-\epsilon_i)\left\|\widetilde{\chi}_i\right\|^2 \end{aligned}, \tag{54}$$

where $0 < \epsilon_i < 1$, and $\left\|\widetilde{\chi}_i\right\| \geq \frac{2\|\mathcal{W}_i \mathcal{Q}_i\|\|h_i\|}{\epsilon_i}$. According to Assumption 1, one derives:

$$\left\|\widetilde{\chi}_i\right\|(t) \leq \max\left\{ \Omega\left(\left\|\widetilde{\chi}_i(0)\right\|, t\right), \Xi(\|h_i\|) \right\}, \tag{55}$$

where $\Omega(\bullet)$ is the $\mathcal{KL}$ function, and $\Xi(\bullet)$ is the $\mathcal{K}_\infty$ function defined as:

$$\Xi(\|h_i\|) = \sqrt{\frac{\lambda_{max}(\mathcal{W}_i)}{\lambda_{min}(\mathcal{W}_i)}} \frac{2\|\mathcal{W}_i \mathcal{Q}_i\|}{\epsilon_i} \|h_i\|. \tag{56}$$

Hence, the estimation dynamics are ISS with respect to $h_i$.

Consider the following Lyapunov function:

$$V_4^i = \frac{1}{2}\left(\varkappa_3^i\right)^T \varkappa_3^i + \frac{1}{2}\left(e_5^i\right)^T e_5^i + \frac{1}{2}\left(e_6^i\right)^T e_6^i. \tag{57}$$

The derivative of (55) is:

$$\begin{aligned} \dot{V}_4^i &= \left(\varkappa_3^i\right)^T \dot{\varkappa}_3^i + \left(e_5^i\right)^T \dot{e}_5^i + \left(e_6^i\right)^T \dot{e}_6^i \\ &\leq -\left(K_1^i - \tfrac{1}{2}I\right)\left(\varkappa_3^i\right)^T \varkappa_3^i - \left(K_2^i - \tfrac{1}{2}I\right)\left(e_5^i\right)^T e_5^i - \left(\tfrac{1}{T_\ell^i} - I\right)\left(e_6^i\right)^T e_6^i + \Theta_i \end{aligned}, \tag{58}$$

where $\Theta_i = \frac{\dot{\varsigma}_i^T \dot{\varsigma}_i}{2} + \frac{\widetilde{\sigma}_i^T \widetilde{\sigma}_i}{2}$. Fully actuated vessels in practical applications have bounded control inputs and velocities. The control input $\varsigma_i$ and its derivation are a continuous and bounded function, ensuring precise and stable maneuvers while adhering to operational standards. The observer error $\widetilde{\sigma}_i$ is bounded from (50)–(56). Hence, $\Theta_i$ is bounded. The selection of parameters is as follows: $\lambda_{min}\left(K_1^i - \tfrac{1}{2}I\right) > 0$, $\lambda_{min}\left(K_2^i - \tfrac{1}{2}I\right) > 0$, and $\lambda_{min}\left(\frac{1}{T_\ell^i} - I\right) > 0$. Finally, (58) becomes:

$$\dot{V}_4^i \leq -2\zeta_i V_4^i + \Theta_i, \tag{59}$$

where $\zeta_i = \min\left\{ \lambda_{min}\left(K_1^i - \tfrac{1}{2}I\right), \lambda_{min}\left(K_{2,i} - \tfrac{1}{2}I\right), \lambda_{min}(\frac{1}{T_\ell^i} - 1) \right\}$. By solving Equation (59), one derives:

$$0 \leq V_4^i(t) \leq \frac{\Theta_i}{2\zeta_i} + \left( V_4^i(0) - \frac{\Theta_i}{2\zeta_i} \right)e^{-2\zeta_i t}, \tag{60}$$

which implies that $\lim_{t\to\infty} V_4^i(t) = \frac{\Theta_i}{2\zeta_i}$ and all the signals of the closed-loop system are bounded. □

## 5. Simulation

In this section, an empirical evaluation is conducted to assess the effectiveness of the controllers and observers proposed in Section 3. Our study focused on a heterogeneous formation system comprising three underactuated and four fully actuated vessels [38,39].

Table 1 presents the model parameters for the leader $i$, all expressed in the International System of Units (SI). It is noteworthy that both fully actuated vessels and underactuated vessels share the same model parameters. The simulation experiments were conducted using MATLAB 2020a, with a sampling time selected as 0.01, and the integration of the differential Equations was performed using the Runge–Kutta method. The three leaders' initial position and velocity vectors were set as $\eta_1 = [14, 381, 0]^T$, $\eta_2 = [3, 382, 0]^T$, $\eta_3 = [0, 359, 0]^T$, $\nu_1 = [0, 0, 0]^T$, $\nu_2 = [0, 0, 0]^T$, and $\nu_3 = [0, 0, 0]^T$. The four followers' initial position and velocity vectors were set as $\eta_4 = [-20, 380, 0]^T$, $\eta_5 = [-10, 360, 0]^T$, $\eta_6 = [-20, 380, 0]^T$, $\eta_7 = [-40, 378, 0]^T$, $\nu_4 = [0, 0, 0]^T$, $\nu_5 = [0, 0, 0]^T$, $\nu_6 = [0, 0, 0]^T$, and $\nu_7 = [0, 0, 0]^T$. The corresponding equilibrium matrix are as follows:

$$\Pi = \begin{bmatrix} 0.2741 & -0.2741 & -0.2741 & 0.1370 & 0.1370 & 0 & 0 \\ -0.2741 & 0.6852 & 0 & -0.5482 & 0 & 0 & 0.1370 \\ -0.2741 & 0 & 0.6852 & 0 & -0.5482 & 0.1370 & 0 \\ 0.1370 & -0.5482 & 0 & 0.7537 & -0.0685 & -0.2741 & 0 \\ 0.1370 & 0 & -0.5482 & -0.0685 & 0.7537 & 0 & -0.2741 \\ 0 & 0 & 0.1370 & -0.2741 & 0 & 0.2741 & -0.1370 \\ 0 & 0.1370 & -0 & 0 & -0.2741 & -0.1370 & 0.2741 \end{bmatrix} \quad (61)$$

**Table 1.** Parameters.

| Entry | Value | Entry | Value |
|---|---|---|---|
| $m_{11}^i$ | 25.8 | $d_{11}^i$ | $0.72 + 1.33\lvert u_i \rvert + 5.87u_i^2$ |
| $m_{22}^i$ | 33.8 | $d_{22}^i$ | $0.8896 + 36.5\lvert v_i \rvert + 0.805\lvert r_i \rvert$ |
| $m_{23}^i$ | 1.0115 | $d_{23}^i$ | $7.25 + 0.845\lvert v_i \rvert + 3.45\lvert r_i \rvert$ |
| $m_{32}^i$ | 1.0948 | $d_{32}^i$ | $0.0313 + 3.96\lvert v_i \rvert + 0.13\lvert r_i \rvert$ |
| $m_{33}^i$ | 2.76 | $d_{33}^i$ | $1.9 - 0.08\lvert v_i \rvert + 0.75\lvert r_i \rvert$ |
| $c_{13}^i$ | $-33.8v_i - 1.0115r_i$ | $g_u^i$ | $0.0279u_i v_i^2 + 0.0342v_i^3 r_i$ |
| $c_{31}^i$ | $-c_{13}^i$ | $\tau_u^i$ | $2\sin\left(0.08\pi t - \pi/6\right) + 0.5\cos(0.05\pi t + \pi/5)$ |
| $c_{23}^i$ | $25.8u_i$ | $\tau_v^i$ | $2\sin\left(0.08\pi t - \pi/4\right) + 0.5\cos(0.05\pi t + \pi/4)$ |
| $c_{32}^i$ | $-c_{23}^i$ | $\tau_r^i$ | $2\sin\left(0.08\pi t - \pi/5\right) + 0.5\cos(0.05\pi t + \pi/3)$ |
| $g_v^i$ | $0.0912u_i^2 v_i$ | $g_v^i$ | $0.0156u_i r_i^3 + 0.0278u_i v_i^3 r_i$ |

The controller parameters were set as $k_u^1 = k_u^2 = k_u^3 = k_u^4 = 3$, $k_r^1 = k_r^2 = k_r^3 = k_r^4 = 2$, $\hbar_u^1 = \hbar_u^2 = \hbar_u^3 = \hbar_u^4 = 1$, $\hbar_v^1 = \hbar_v^2 = \hbar_v^3 = \hbar_v^4 = 3$, $T_u^1 = T_u^2 = T_u^3 = T_u^4 = 0.05$, $T_r^1 = T_r^2 = T_r^3 = T_r^4 = 0.05$; $K_1^1 = K_1^2 = K_1^3 = K_1^4 = diag\{1, 1, 1\}$, $K_2^1 = K_2^2 = K_2^3 = K_2^4 = diag\{2, 2, 2\}$, and $T_\ell^1 = T_\ell^2 = T_\ell^3 = T_\ell^4 = diag\{0.05, 0.05, 0.05\}$. By employing YALMIP with the sedumi solver to solve (24) and (35), one derives:

$$\mathcal{W}_i = \mathcal{P}_i = \begin{bmatrix} 5.2780 & 0 & 0 & -1.5951 & 0 & 0 \\ 0 & 5.2780 & 0 & 0 & -1.5951 & 0 \\ 0 & 0 & 5.2780 & 0 & 0 & -1.5951 \\ -1.5951 & 0 & 0 & 2.6363 & 0 & 0 \\ 0 & -1.5951 & 0 & 0 & 2.6363 & 0 \\ 0 & 0 & -1.5951 & 0 & 0 & 2.6363 \end{bmatrix} \quad (62)$$

$$\Phi_i = \Upsilon_i = \begin{bmatrix} -2.6547 & 0 & 0 \\ 0 & -2.6547 & 0 \\ 0 & 0 & -2.6547 \\ -5.2780 & 0 & 0 \\ 0 & -5.2780 & 0 \\ 0 & 0 & -5.2780 \end{bmatrix} \quad (63)$$

Figures 2–13 present the simulations, which serve to confirm the effectiveness of the proposed control scheme. In Figure 2, channels in the near-sea environment are

simulated by utilizing the gaps between grey obstacles. The heterogeneous formation system demonstrated its capability to perform various maneuvering operations, including translation, scaling, rotation, and shearing at specific time instances: 148 s, 280 s, 355 s, and 450 s, respectively. Based on the information shown in Figures 3 and 4, it was evident that both the leaders and the followers could precisely track their intended positions. From these figures, it can be observed that under the presence of uncertain models and external disturbances, the tracking errors of the unmanned vessels are consistently and ultimately bounded, aligning with Theorems 1 and 2. Figure 5 showcases the velocities of the three leaders while their corresponding forces and moments are depicted in Figure 8. On the other hand, Figures 6 and 7 display the velocities of the four followers, and their complementary forces and moments are illustrated in Figures 9 and 10. Figures 8–10 show that the leaders' and followers' forward thrust and turning moment are both bounded. The lumped disturbances experienced by the unmanned vessels were effectively captured through the observation estimation designed in this section, as shown in Figures 11–13.

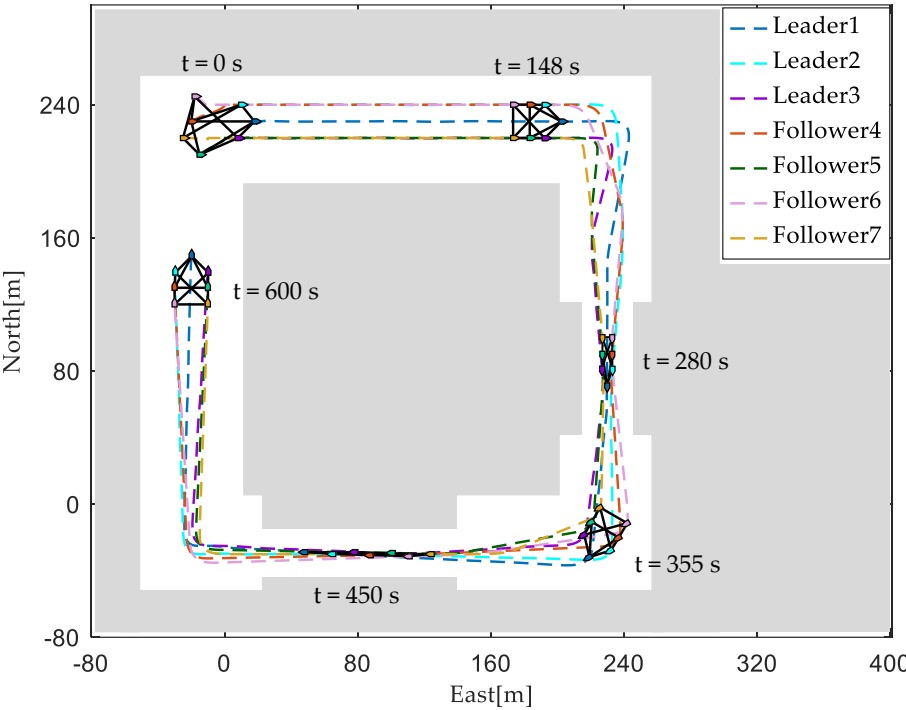

**Figure 2.** The formation maneuver trajectories of the heterogeneous formation system.

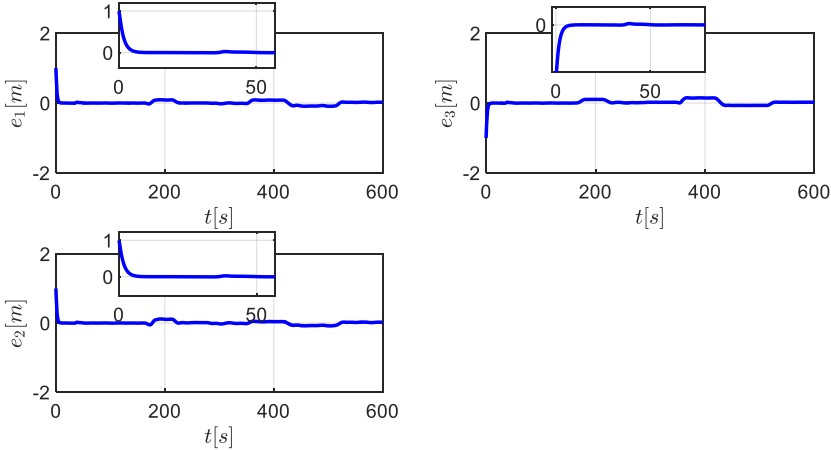

**Figure 3.** The tracking error of the leaders.

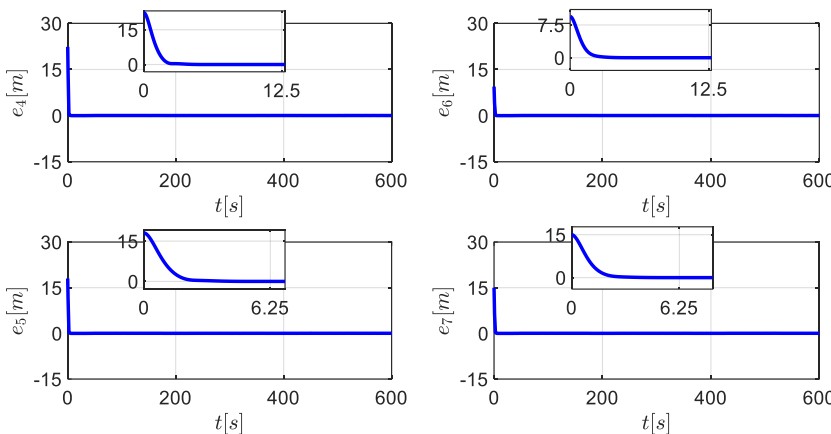

**Figure 4.** The tracking error of the followers.

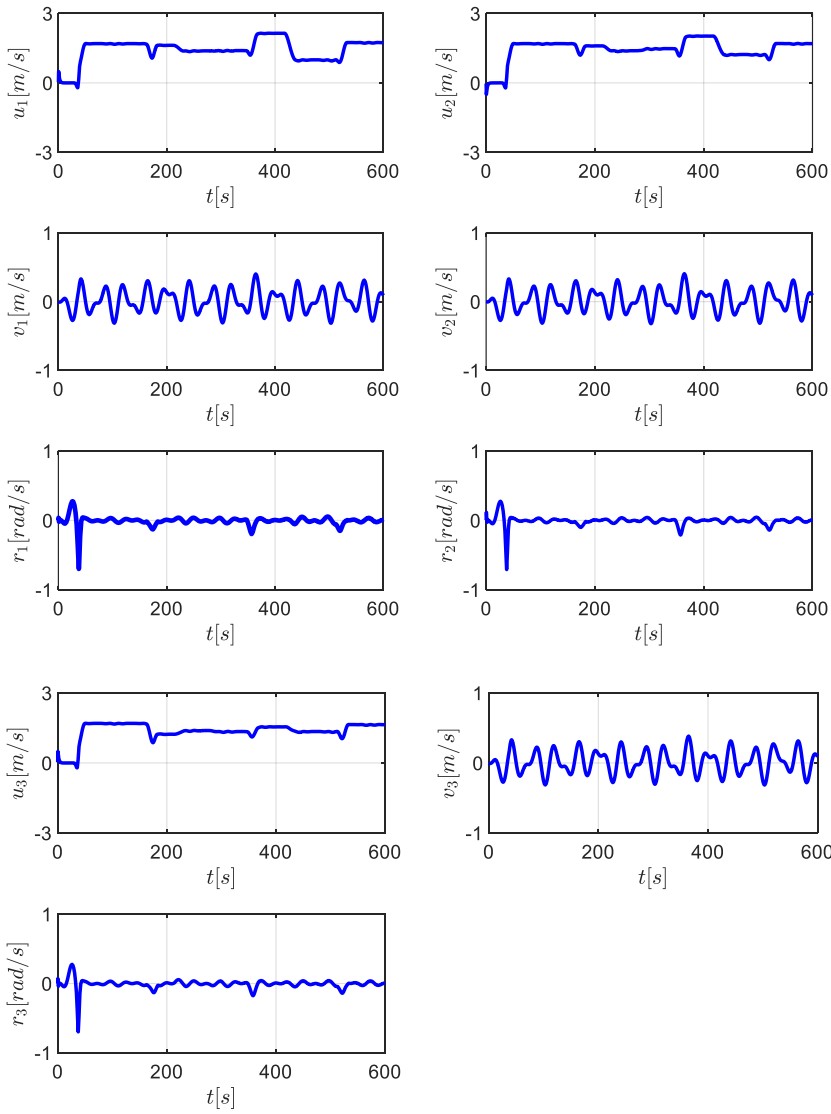

**Figure 5.** The velocities of leader 1, leader 2, and leader 3.



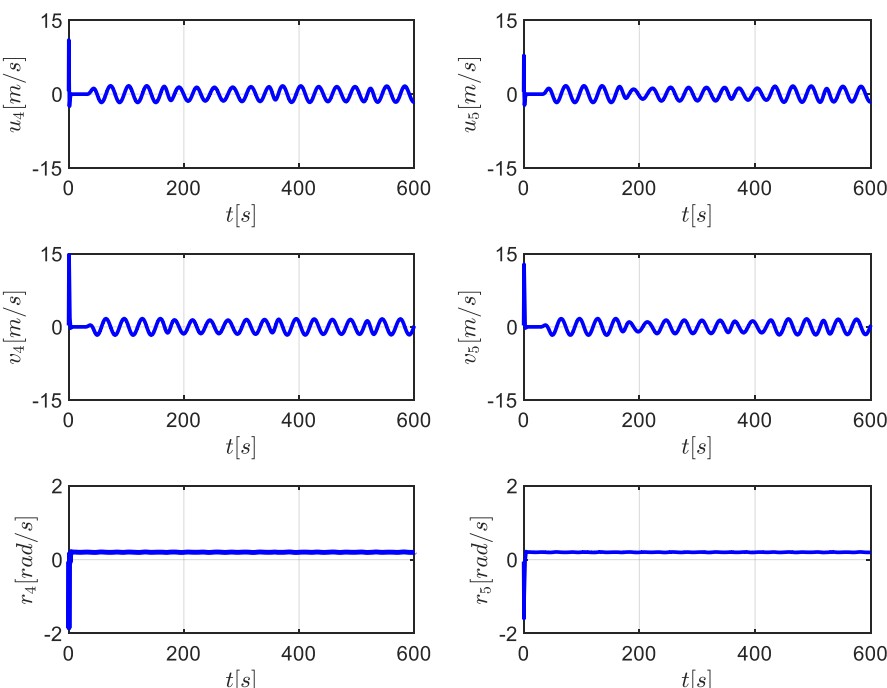

**Figure 6.** The velocities of follower 4 and follower 5.

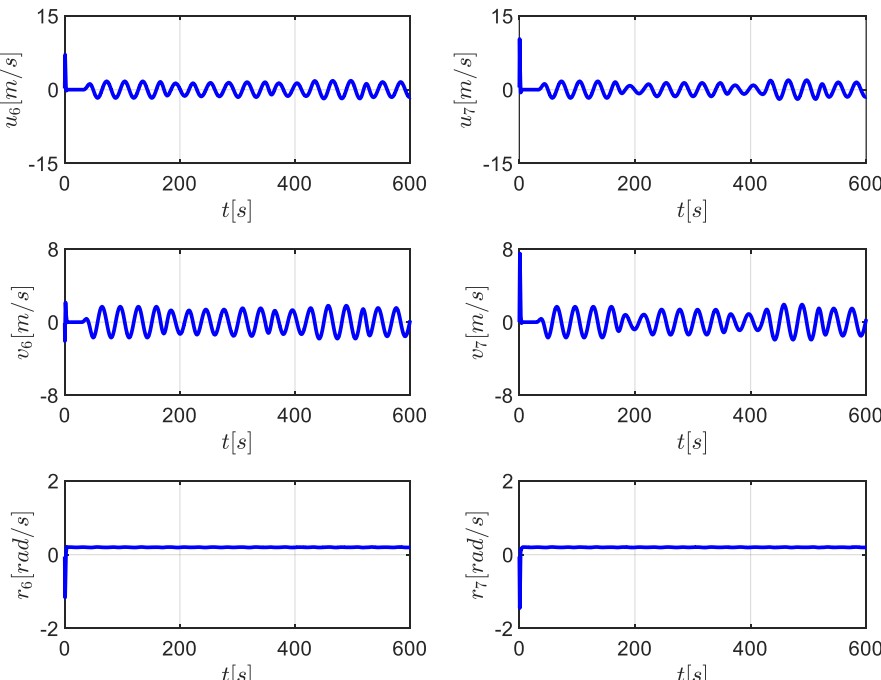

**Figure 7.** The velocities of the follower 6 and follower 7.

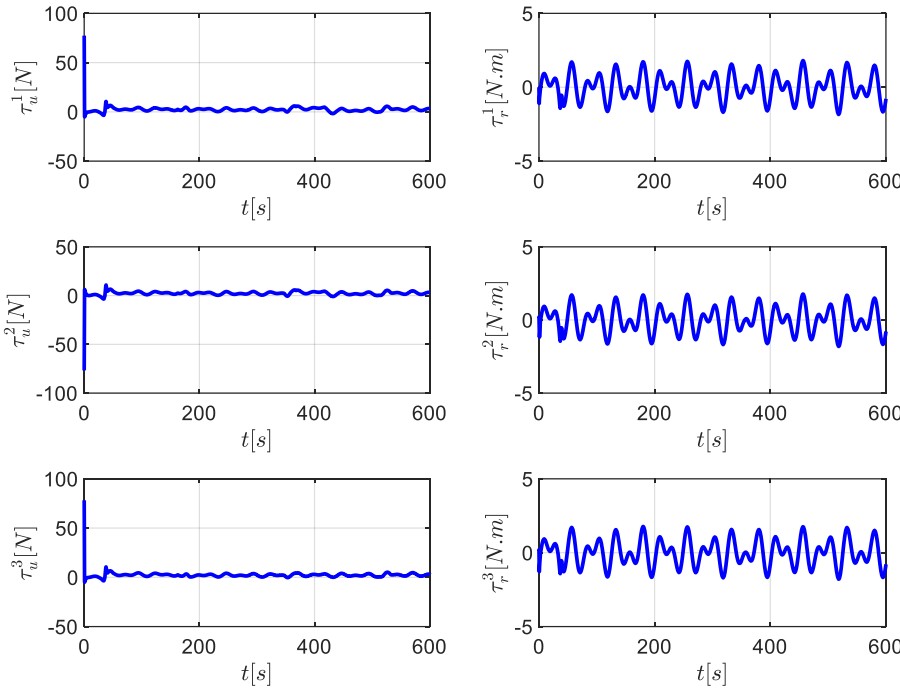

**Figure 8.** The forces and moments of the leaders.

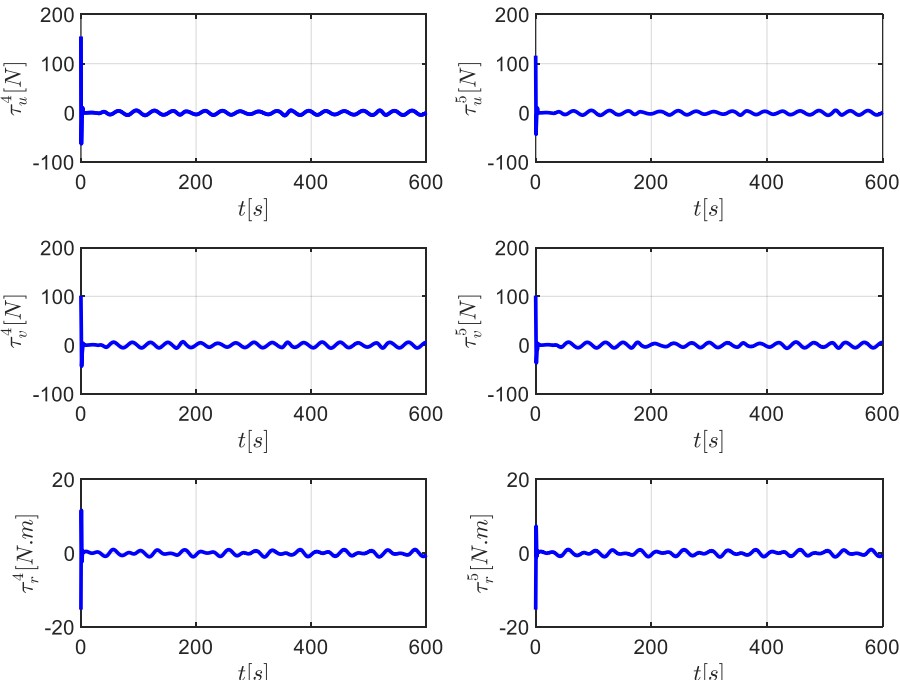

**Figure 9.** The forces and moments of follower 4 and follower 5.

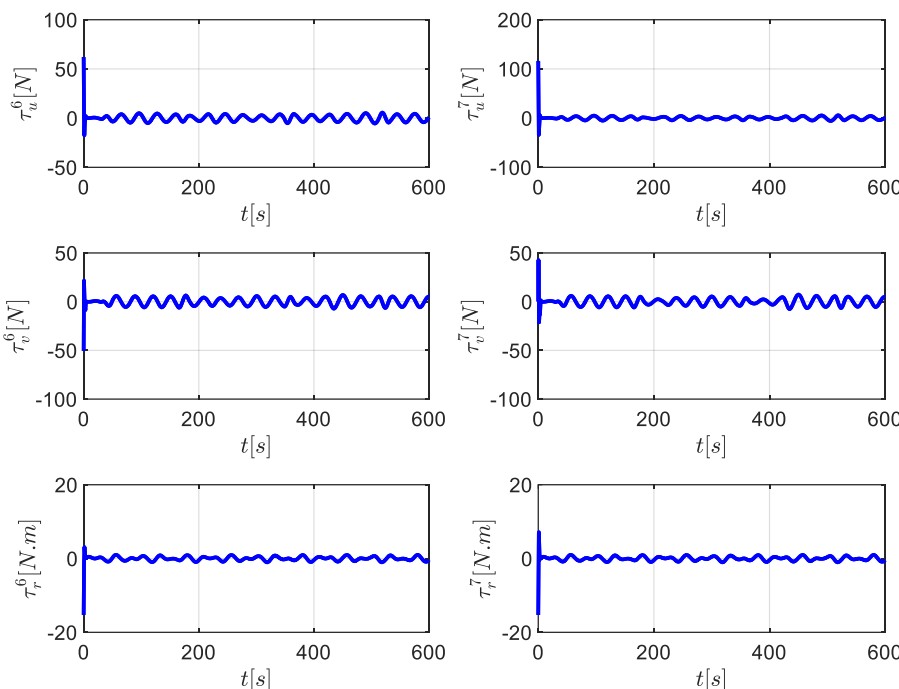

**Figure 10.** The forces and moments of follower 6 and follower 7.

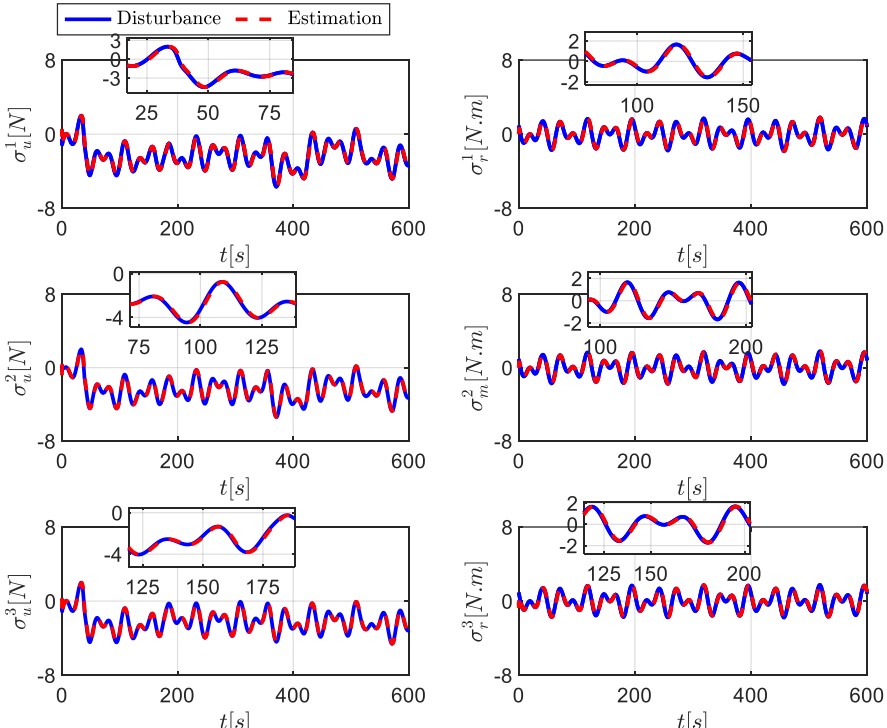

**Figure 11.** The lumped disturbance estimation of the leaders with LMI.

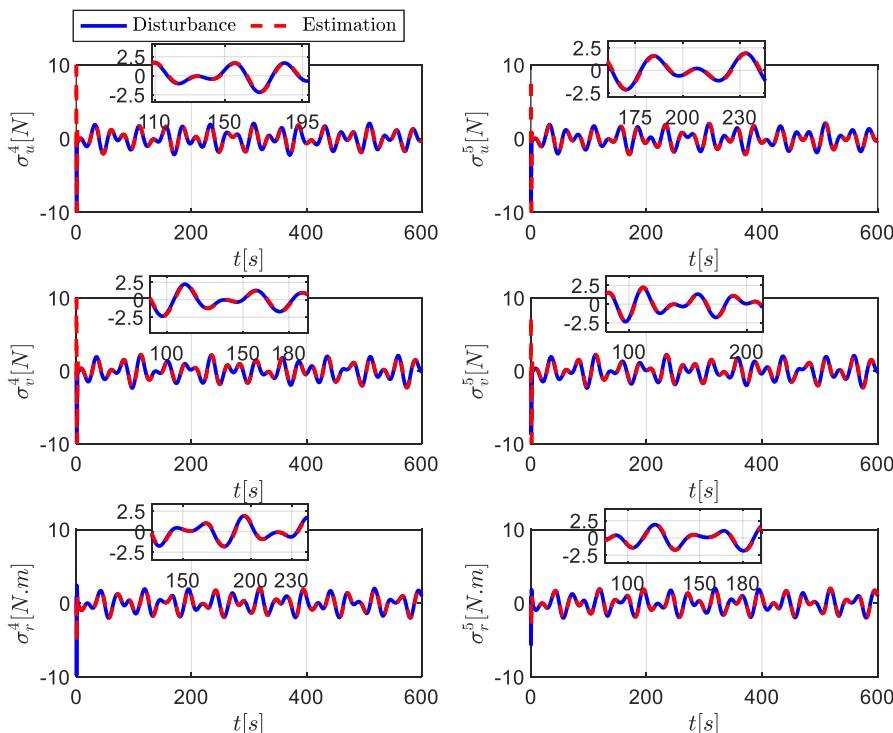

**Figure 12.** The lumped disturbance estimation of follower 4 and follower 5 with LMI.

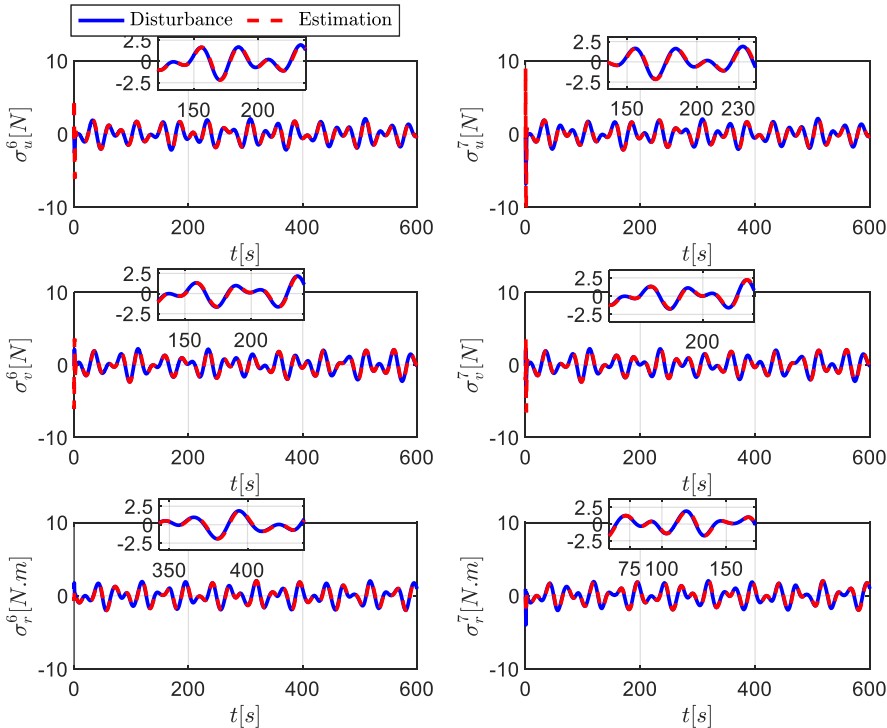

**Figure 13.** The lumped disturbance estimation of follower 6 and follower 7 with LMI.

To validate the estimation performance of the observer optimized through LMI (Linear Matrix Inequality), this section compared the gains obtained through LMI solutions with those obtained based on Lyapunov stability conditions. The observer gain without LMI was set as: $\mathcal{H}_1 = \mathcal{H}_2 = \mathcal{H}_3 = [diag(10,5,5), diag(5.5,5.5,3)]^T$, $\mathcal{H}_4 = \mathcal{H}_5 = \mathcal{H}_6 = \mathcal{H}_7 = [diag(5,5,2), diag(11,2,2)]^T$. Figures 14–16 present the observation error obtained by solving the observer gain using LMI and directly setting the observer gain. Those figures

show that the gains obtained using the Linear Matrix Inequality (LMI) approach yield higher accuracy in estimating the aggregate disturbance than gains computed based on Lyapunov stability conditions. This indicated that the optimization criterion (24) improved the observer's performance and results in more accurate state estimation. These simulation results proved the control scheme's effectiveness in achieving desired formation maneuvers.

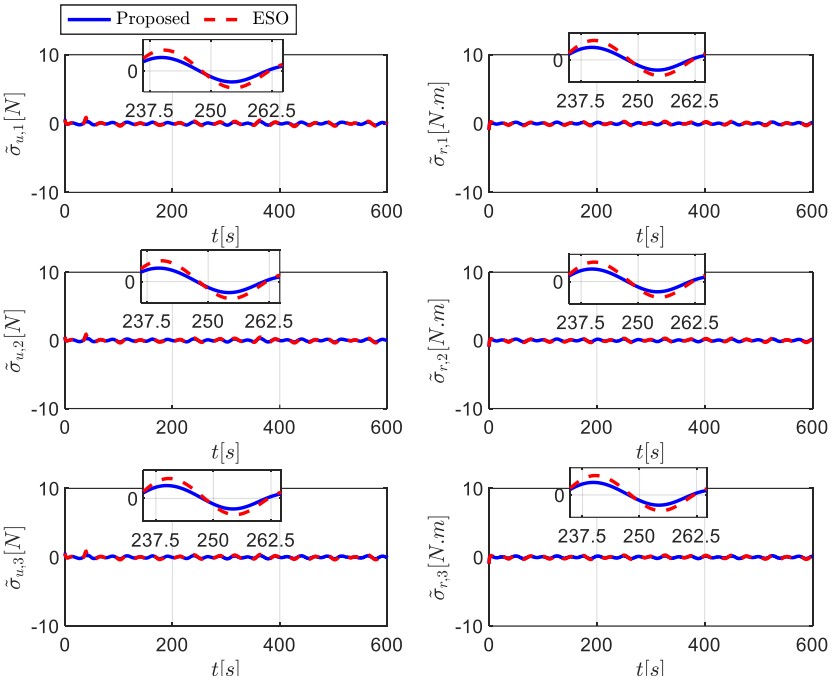

**Figure 14.** The estimation error of follower 1 and follower 2.

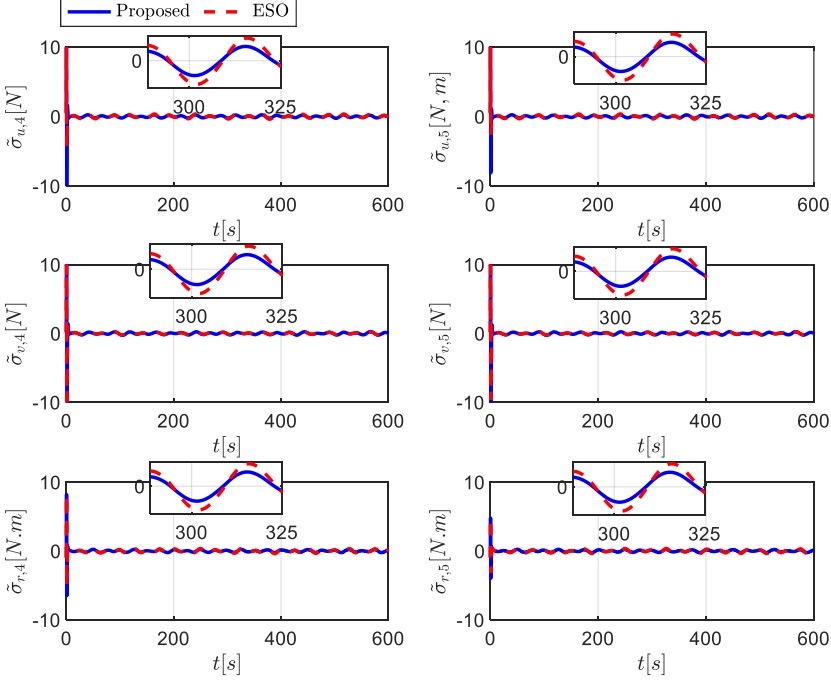

**Figure 15.** The estimation error of follower 4 and follower 5.

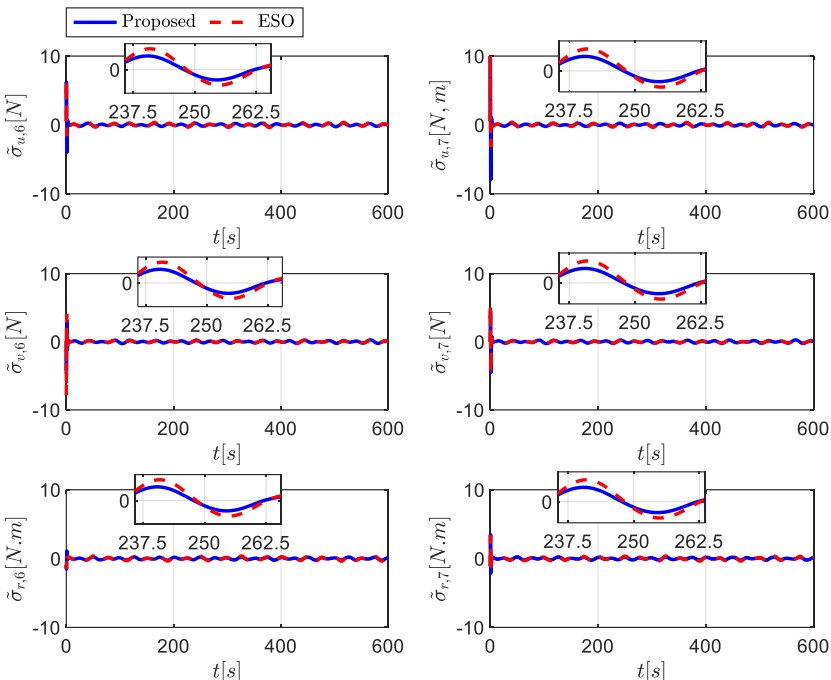

**Figure 16.** The estimation error of follower 6 and follower 7.

## 6. Conclusions

This paper delves into affine formation maneuver control for multi-heterogeneous unmanned surface vessels (USV) in near-sea environments. The proposed control scheme offers a comprehensive solution to tackle the complexities of navigating through narrow channels while upholding the integrity of the formation. By incorporating affine transformations, the formation system gains the ability to adapt its configuration, boosting the overall flexibility and versatility of the USVs. Furthermore, the anti-perturbation formation tracking controller ensures precise and accurate tracking of arbitrary formation transformations by fully actuated and underactuated vessels. This capacity empowers the formation to swiftly adapt to evolving mission requirements and dynamic environmental conditions. As a future research direction, this study can be extended to explore collision avoidance strategies involving dynamic obstacles and effectively address the challenges tied to input saturation. These efforts will significantly fortify the formations' maneuverability, especially when navigating intricate and demanding maritime environments, where precise control and adaptability are crucial. Additionally, a deeper investigation into integrating real-time adaptive algorithms for collision avoidance and developing advanced control mechanisms could enhance the formations' responsiveness and resilience, enabling them to excel in dynamic and uncertain operational scenarios.

**Author Contributions:** Conceptualization, Y.L. and X.L.; methodology, Y.L.; software, Y.L. and C.Z.; validation, Y.L.; formal analysis, X.L.; investigation, Y.L. and C.Z.; resources, X.L.; data curation, Y.L.; writing—original draft preparation, Y.L.; writing—review and editing, X.L. and C.Z.; supervision, X.L. All authors have read and agreed to the published version of the manuscript.

**Funding:** This work was supported by National Natural Science Foundation of China (the funding number is No. 52071102, No. 51679057, No. 51609046, No. 51909044), the Province Science Fund for Distinguished Young Schol-ars (No. J2016JQ0052).

**Institutional Review Board Statement:** Not applicable.

**Informed Consent Statement:** Not applicable.

**Data Availability Statement:** Not applicable.

**Conflicts of Interest:** The authors declare no conflict of interest.

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
