# Peer review of "Affine Formation Maneuver Control for Multi-Heterogeneous Unmanned Surface Vessels in Narrow Channel Environments"

_jmse, doi:10.3390/jmse11091811_

Round 1

Reviewer 1 Report

line 205 blur image - improve quality of figure 1 (higher resolution bitmap or vector text)

line 329 Table 1. Parameters. How did the authors obtain the data for this table? How did you calculate the values of the given parameters used in proposed model? How are all these model and initial parameters verified? Do you have a bibliography to substantiate these simulation models or output data? I strongly suggest that you include an additional bibliography in lines 99 through 397 to support the article for the review process. That is almost 300 lines of your text without any bibliography!

The authors have prepared present bibliography according to the instructions for MDPI authors, but I have some suggestions for improvement:

line 434 please add: DOI: 10.1109/CCDC.2012.6244096, Publisher: IEEE & Conference Location: Taiyuan 

line 437 please add: DOI: 10.1109/CYBER.2018.8688081, Publisher: IEEE, Conference Location: Tianjin, China 

line 440 please add: DOI: 10.23919/ACC.2019.8814890, Publisher: IEEE, Conference Location: Philadelphia, PA, USA  

line 337 to 350 I would like you to improve the presentation of your article by putting the text to the figures next to it (above or below the figures) (lines 336 to 373)

line 9 "we investigate..." look line 11 explanation

line 11 "Firstly, we utilize..." it would be better not to use this first person plural - the personal pronoun we and the related pronouns us, ours, and ourselves are all first-person. I suggest authors to use third-person point of view "it is utilized..." (or even better instead of utilized use word used)

line 12 "Secondly, we introduce..." I suggest e.g. "Secondly in this article is introduced..."

line 15 is well done "This paper utilizes..."

line 32, 38, 117, 119, 141, 176, 209, 232, 244, 258, 274, 277, 278, 282, 285, 296, 299, 300, 308, 317, 333, 338 I suggest avoiding the word "we".

Reviewer 2 Report

1. Introduction can be  improved by incorporating the recent results of formation control problem of these papers:

https://doi.org/10.1016/j.ifacol.2023.01.153.

https://ieeexplore.ieee.org/document/10178289.

https://doi.org/10.1016/j.ifacol.2021.12.029.

https://ieeexplore.ieee.org/abstract/document/9555265.

2. What is novelty and contribution of the paper, present explicitly.

3. References should be in same format.

4. Comparative results should be provided.

Reviewer 3 Report

1. 3 key contributions introduced in Section 1 should be included or at least briefly mentioned in Abstract.

2. For all Assumptions, the corresponding Remarks should be provided for reader's better understanding.

3. Eq. (60) was solved by using YALMIP. However, the solver used was not presented. More detailed saplanation is needed.

4. The “anti-perturbation” is emphasized in this paper. As for how this article resists perturbation, it is not clearly shown or explained.

5. More detailed descriptions for each figure are needed. Explain what each figure represents, what variables or factors are being depicted, etc.

6. In Figures 11 to 13, it appears that the disturbance tracking was done through the estimation too accurately. There is a question about whether such high tracking accuracy is achievable in real experiments. Please explain if this level of accuracy is feasible based on actual experimental results such as real situation.

7. To enhance the credibility of the real-world experiment results, it might be worthwhile to consider conducting simple hardware experiments. Is there a way to include information about such experiments?

The manuscript needs English proofreading. Moreover, it is strongly recommended to take the grammar check-up and language polishing. 

Round 2

Reviewer 1 Report

The authors have implemented all my suggestions for improvement (from Review Round 1). Congratulations on the revision.

Author Response

We sincerely appreciate your feedback and recognition of the revisions made in response to your suggestions. Your encouragement means a lot to us, and we are pleased you found the improvements satisfactory. If you have any further comments or if there are additional ways we can enhance the manuscript, please don't hesitate to inform us. Your feedback has been precious in refining our work, and we hope for your continued support.

Reviewer 3 Report

The overall quality and completeness of the manuscript has been greatly improved compared to the previous submission. 

The manuscript seems to be fine. However, some sentences and expressions still need to be checked again.

Author Response

Thank you for your feedback on the manuscript. We greatly appreciate your input, and we will carefully review the sentences and expressions in the manuscript once again to enhance its overall quality. Your suggestions are precious to us, and we will ensure that the language and words in the paper are more transparent and precise.

Reviewer 4 Report

Thank you for your interesting manuscript and for the comments. In my opinion, the manuscript has been improved. I recommend it for publication after slight revision.

1)     Sorry, I must have been inaccurate in my previous comment on the optimization criterion (24). I did not mean the mathematical meaning of this criterion, but the practical one. How does this affect the performance of the observer? For example, is it the minimization of the estimation time or something else? Please clarify this also in the manuscript itself.

2)     In the Section 5 with simulation results, you have added a comparison of the observer's performance with gains accepted without LMI and using the LMI solution. Please indicate in the manuscript what the values of the gain accepted without LMI were. This is necessary if the reader decides to repeat the experiment.
